# Optimal Transport for Treatment Effect Estimation

**Hao Wang**[1]   **Zhichao Chen**[1]   **Jiajun Fan**[2]   **Haoxuan Li**[3]   **Tianqiao Liu**[4]
**Weiming Liu**[1]   **Quanyu Dai**[5]*   **Yichao Wang**[5]   **Zhenhua Dong**[5]   **Ruiming Tang**[5]
[1]Zhejiang University   [2]Tsinghua University   [3]Peking University
[4]Purdue University   [5] Huawei Noah's Ark Lab
haohaow@zju.edu.cn   quanyu.dai@connect.polyu.hk

## Abstract

Estimating conditional average treatment effect from observational data is highly challenging due to the existence of treatment selection bias. Prevalent methods mitigate this issue by aligning distributions of different treatment groups in the latent space. However, there are two critical problems that these methods fail to address: (1) mini-batch sampling effects (MSE), which causes misalignment in non-ideal mini-batches with outcome imbalance and outliers; (2) unobserved confounder effects (UCE), which results in inaccurate discrepancy calculation due to the neglect of unobserved confounders. To tackle these problems, we propose a principled approach named **E**ntire **S**pace **C**ounter**F**actual **R**egression (ESCFR), which is a new take on optimal transport in the context of causality. Specifically, based on the framework of stochastic optimal transport, we propose a relaxed mass-preserving regularizer to address the MSE issue and design a proximal factual outcome regularizer to handle the UCE issue. Extensive experiments demonstrate that our proposed ESCFR can successfully tackle the treatment selection bias and achieve significantly better performance than state-of-the-art methods.

## 1   Introduction

The estimation of the conditional average treatment effect (CATE) through randomized controlled trials serves as a cornerstone in causal inference, finding applications in diverse fields such as health care [63], e-commerce [2, 43, 81], and education [15]. Although randomized controlled trials are often considered the gold standard for CATE estimation [38, 53], the associated financial and ethical constraints can make them infeasible in practice [39, 79]. Therefore, alternative approaches that rely on observational data have gained prominence. For instance, drug developers could utilize post-marketing monitoring reports to assess the efficacy of new medications rather than conducting costly clinical A/B trials. With the growing access to observational data, estimating CATE from observational data has attracted intense research interest [41, 80, 84].

Estimating CATE with observational data has two main challenges: (1) missing counterfactuals, *i.e.*, only one factual outcome out of all potential outcomes can be observed; (2) treatment selection bias, *i.e.*, individuals have different propensities for treatment, leading to non-random treatment assignments and a resulting covariate shift between the treated and untreated groups [77, 79]. Traditional meta-learners [36] handle the issue of missing counterfactuals by breaking down the CATE estimation into more tractable sub-problems of factual outcome estimation. However, the covariate shift makes it difficult to generalize the factual outcome estimators trained within respective treatment groups to the entire population and thus biases the derived CATE estimator.

Beginning with counterfactual regression [67] and its remarkable performance, there are various attempts that handle the selection bias by minimizing the distribution discrepancy between the

---

*Corresponding author.

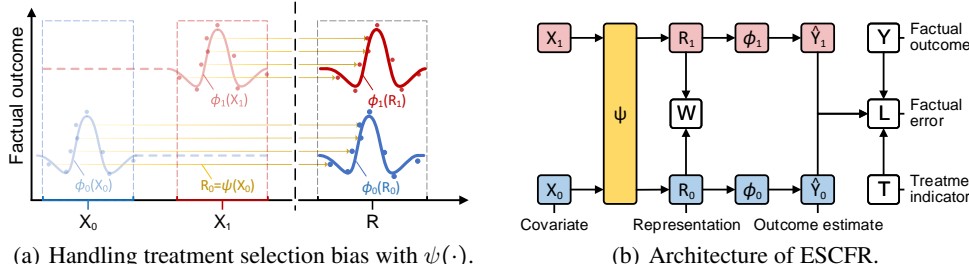

(a) Handling treatment selection bias with $\psi(\cdot)$.

(b) Architecture of ESCFR.

Figure 1: Overview of handling treatment selection bias with ESCFR. The red and blue colors signify the treated and untreated groups, respectively. (a) The treatment selection bias manifests as a distribution shift between $X_1$ and $X_0$. The scatters and curves represent the units and the fitted outcome mappings. (b) ESCFR mitigates the selection bias by aligning units from different treatment groups in the representation space: $R = \psi(X)$, which enables $\phi_1$ and $\phi_0$ to generalize across groups.

treatment groups in the representation space [13, 27, 85, 86]. However, two critical issues with these methods have long been neglected, which impedes them from handling the treatment selection bias. The first issue is the mini-batch sampling effects (MSE). Specifically, current representation-based methods compute the distribution discrepancy within mini-batches instead of the entire data space, making it vulnerable to bad sampling cases. For example, the presence of an outlier in a mini-batch can falsely increase the estimate of the distribution discrepancy, thereby misleading the update of the estimator. The second issue is the unobserved confounder effects (UCE). Specifically, current approaches mainly operate under the unconfoundedness assumption [48] and ignore the pervasive influence of unobserved confounders, which makes the resulting estimators biased given the existence of unobserved confounders.

**Contributions and outline.** In this paper, we propose an effective CATE estimator based on optimal transport, namely Entire Space CounterFactual Regression (ESCFR), which handles both the MSE and UCE issues with a generalized sinkhorn discrepancy. Specifically, after presenting preliminaries in Section 2, we redefine the problem of CATE estimation within the framework of stochastic optimal transport in Section 3.1. We subsequently showcase the MSE issue faced by existing approaches in Section 3.2 and introduce a mass-preserving regularizer to counter this issue. In Section 3.3, we explore the UCE issue and propose a proximal factual outcome regularizer to mitigate its impact. The architecture and learning objectives of ESCFR are elaborated upon in Section 3.4, followed by the presentation of our experimental results in Section 4.

## 2 Preliminaries

### 2.1 Causal inference from observational data

This section formulates preliminaries in observational causal inference. We first formalize the fundamental elements in Definition 2.1 following the general notation convention[2].

**Definition 2.1.** *Let $X$ be the random variable of covariates, with support $\mathcal{X}$ and distribution $\mathbb{P}(x)$; Let $R$ be the random variable of induced representations, with support $\mathcal{R}$ and distribution $\mathbb{P}(r)$; Let $Y$ be the random variable of outcomes, with support $\mathcal{Y}$ and distribution $\mathbb{P}(y)$; Let $T$ be the random variable of treatment indicator, with support $\mathcal{T} = \{0, 1\}$ and distribution $\mathbb{P}(T)$.*

In the potential outcome framework [61], a unit characterized by the covariates $x$ has two potential outcomes, namely $Y_1(x)$ given it is treated and $Y_0(x)$ otherwise. The CATE is defined as the conditionally expected difference of potential outcomes as follow:

$$\tau(x) := \mathbb{E}\left[Y_1 - Y_0 \mid x\right], \tag{1}$$

---

[2]We use uppercase letters, *e.g.*, $X$ to denote a random variable, and lowercase letters, *e.g.*, $x$ to denote an associated specific value. Letters in calligraphic font, *e.g.*, $\mathcal{X}$ represent the support of the corresponding random variable, and $\mathbb{P}()$ represents the probability distribution of the random variable, *e.g.*, $\mathbb{P}(X)$.

where one of these two outcomes is always unobserved in the collected data. To address such missing counterfactuals, the CATE estimation task is commonly decomposed into outcome estimation subproblems that are solvable with supervised learning method [36]. For example, T-learner models the factual outcomes $Y$ for units in the treated and untreated groups separately; S-learner regards $T$ as a covariate, and then models $Y$ for units in all groups simultaneously. The CATE estimate is then the difference of the estimated outcomes when $T$ is set to treated and untreated.

**Definition 2.2.** *Let $\psi : \mathcal{X} \to \mathcal{R}$ be a mapping from support $\mathcal{X}$ to $\mathcal{R}$, i.e., $\forall x \in \mathcal{X}$, $r = \psi(x) \in \mathcal{R}$. Let $\phi : \mathcal{R} \times \mathcal{T} \to \mathcal{Y}$ be a mapping from support $\mathcal{R} \times \mathcal{T}$ to $\mathcal{Y}$, i.e., it maps the representations and treatment indicator to the corresponding factual outcome. Denote $Y_1 = \phi_1(R)$ and $Y_0 = \phi_0(R)$, where we abbreviate $\phi(R, T = 1)$ and $\phi(R, T = 0)$ to $\phi_1(R)$ and $\phi_0(R)$, respectively, for brevity.*

TARNet [67] combines T-learner and S-learner to achieve better performance, which consists of a representation mapping $\psi$ and an outcome mapping $\phi$ as defined in Definition 2.2. For an unit with covariates $x$, TARNet estimates the CATE as:

$$\hat{\tau}_{\psi,\phi}(x) = \hat{Y}_1 - \hat{Y}_0, \quad \hat{Y}_1 = \phi_1(\psi(x)), \quad \hat{Y}_0 = \phi_0(\psi(x)), \tag{2}$$

where $\psi$ is trained over all individuals, $\phi_1$ and $\phi_0$ are trained over the treated and untreated group, respectively. Finally, the quality of CATE estimation is evaluated with the precision in estimation of heterogeneous effect (PEHE) metric:

$$\epsilon_{\mathrm{PEHE}}(\psi,\phi) \coloneqq \int_{\mathcal{X}} \left(\hat{\tau}_{\psi,\phi}(x) - \tau(x)\right)^2 \mathbb{P}(x)\, dx. \tag{3}$$

However, according to Figure 1(a), the treatment selection bias causes a distribution shift of covariates across groups, which misleads $\phi_1$ and $\phi_0$ to overfit their respective group's properties and generalize poorly to the entire population. Therefore, the CATE estimate $\hat{\tau}$ by these methods would be biased.

## 2.2 Discrete optimal transport and Sinkhorn divergence

Optimal transport (OT) quantifies distribution discrepancy as the minimum transport cost, offering a tool to quantify the selection bias in Figure 1(a). Monge [49] first formulated OT as finding an optimal mapping between two distributions. Kantorovich [34] further proposed a more applicable formulation in Definition 2.3, which can be seen as a generalization of the Monge problem.

**Definition 2.3.** *For empirical distributions $\alpha$ and $\beta$ with $n$ and $m$ units, respectively, the Kantorovich problem aims to find a feasible plan $\pi \in \mathbb{R}_+^{n \times m}$ which transports $\alpha$ to $\beta$ at the minimum cost:*

$$\mathbb{W}(\alpha,\beta) \coloneqq \min_{\boldsymbol{\pi} \in \Pi(\alpha,\beta)} \langle \mathbf{D}, \boldsymbol{\pi} \rangle, \ \Pi(\alpha,\beta) \coloneqq \left\{ \boldsymbol{\pi} \in \mathbb{R}_+^{n \times m} : \boldsymbol{\pi} \mathbf{1}_m = \mathbf{a}, \boldsymbol{\pi}^{\mathrm{T}} \mathbf{1}_n = \mathbf{b} \right\}, \tag{4}$$

*where $\mathbb{W}(\alpha,\beta) \in \mathbb{R}$ is the Wasserstein discrepancy between $\alpha$ and $\beta$; $\mathbf{D} \in \mathbb{R}_+^{n \times m}$ is the unit-wise distance between $\alpha$ and $\beta$, which is implemented with the squared Euclidean distance; $\mathbf{a}$ and $\mathbf{b}$ indicate the mass of units in $\alpha$ and $\beta$, and $\Pi$ is the feasible transportation plan set which ensures the mass-preserving constraint holds.*

Since exact solutions to (4) often come with high computational costs [5], researchers would commonly add an entropic regularization to the Kantorovich problem as follow:

$$\mathbb{W}^{\epsilon}(\alpha,\beta) \coloneqq \langle \mathbf{D}, \boldsymbol{\pi}^{\epsilon} \rangle, \ \boldsymbol{\pi}^{\epsilon} \coloneqq \arg\min_{\boldsymbol{\pi} \in \Pi(\alpha,\beta)} \langle \mathbf{D}, \boldsymbol{\pi} \rangle - \epsilon \mathrm{H}(\boldsymbol{\pi}), \ \mathrm{H}(\boldsymbol{\pi}) \coloneqq - \sum_{i,j} \boldsymbol{\pi}_{i,j} \left(\log(\boldsymbol{\pi}_{i,j}) - 1\right), \tag{5}$$

which makes the problem $\epsilon$-convex and solvable with the Sinkhorn algorithm [19]. The Sinkhorn algorithm only consists of matrix-vector products, making it suited to be accelerated with GPUs.

## 3 Proposed method

In this section, we present the Entire Space CounterFactual Regression (ESCFR) approach, which leverages optimal transport to tackle the treatment selection bias. We first illustrate the stochastic optimal transport framework for distribution discrepancy quantification across treatment groups, and demonstrate its efficacy for improving the performance of CATE estimators. Based on the framework, we then propose a relaxed mass-preserving regularizer to address the sampling effect, and a proximal factual outcome regularizer to handle the unobserved confounders. We finally open a new thread to summarize the model architecture, learning objectives, and optimization algorithm.

## 3.1 Stochastic optimal transport for counterfactual regression

Representation-based approaches mitigate the treatment selection bias by calculating distribution discrepancy in the representation space and then minimizing it. OT is a preferred method to quantify the discrepancy due to its numerical advantages and flexibility over competitors. It is numerically stable in the cases where other methods, such as $\phi$-divergence (*e.g.*, Kullback-Leibler divergence), fails [64]. Compared with adversarial discrepancy measures [7, 33, 87], the it can be calculated efficiently and integrated naturally with the traditional supervised learning framework.

Formally, we denote the OT discrepancy between treatment groups as $\mathbb{W}\left(\mathbb{P}_\psi^{T=1}(r), \mathbb{P}_\psi^{T=0}(r)\right)$, where $\mathbb{P}_\psi^{T=1}(r)$ and $\mathbb{P}_\psi^{T=0}(r)$ are the distributions of representations in treated and untreated groups, respectively, induced by the mapping $r = \psi(x)$. The discrepancy is differentiable with respect to $\psi$ [24], and thus can be minimized by updating the representation mapping $\psi$ with gradient-based optimizers.

**Definition 3.1.** *Let $\hat{\mathbb{P}}^{T=1}(x) := \{x_i^{T=1}\}_{i=1}^n$ and $\hat{\mathbb{P}}^{T=0}(x) := \{x_i^{T=0}\}_{i=1}^m$ be the empirical distributions of covariates at a mini-batch level, which contain $n$ treated units and $m$ untreated units, respectively; $\hat{\mathbb{P}}_\psi^{T=1}(r)$ and $\hat{\mathbb{P}}_\psi^{T=0}(r)$ be that of representations induced by the mapping $r = \psi(x)$ in Definition 2.2.*

However, since prevalent neural estimators mainly update parameters with stochastic gradient methods, only a fraction of the units is accessible within each iteration. A shortcut in this context is to calculate the discrepancy at a stochastic mini-batch level:

$$\hat{\mathbb{W}}_\psi := \mathbb{W}\left(\hat{\mathbb{P}}_\psi^{T=1}(r), \hat{\mathbb{P}}_\psi^{T=0}(r)\right). \tag{6}$$

The effectiveness of this shortcut is investigated by Theorem 3.1 (refer to Appendix A.4 for proof), which demonstrates that the PEHE can be optimized by iteratively minimizing the estimation error of factual outcomes and the mini-batch group discrepancy in (6).

**Theorem 3.1.** *Let $\psi$ and $\phi$ be the representation mapping and factual outcome mapping, respectively; $\hat{\mathbb{W}}_\psi$ be the group discrepancy at a mini-batch level. With the probability of at least $1 - \delta$, we have:*

$$\epsilon_{\text{PEHE}}(\psi, \phi) \leq 2[\epsilon_F^{T=1}(\psi, \phi) + \epsilon_F^{T=0}(\psi, \phi) + B_\psi \hat{\mathbb{W}}_\psi - 2\sigma_Y^2 + \mathcal{O}(\frac{1}{\delta N})], \tag{7}$$

*where $\epsilon_F^{T=1}$ and $\epsilon_F^{T=0}$ are the expected errors of factual outcome estimation, $N$ is the batch size, $\sigma_Y^2$ is the variance of outcomes, $B_\psi$ is a constant term, and $\mathcal{O}(\cdot)$ is a sampling complexity term.*

## 3.2 Relaxed mass-preserving regularizer for sampling effect

Although Theorem 3.1 guarantees that the empirical OT discrepancy (6) bounds the PEHE, the sampling complexity term $\mathcal{O}(\cdot)$ inspires us to investigate the potential risks of bad cases caused by stochastic sampling. Precisely, the term $\mathcal{O}(\cdot)$ results from the discrepancy between the entire population and the sampled mini-batch units (see (30) and (32) in Appendix A) which is highly dependent on the uncontrollable sampling quality. Therefore, a reliable discrepancy measure should be robust to bad sampling cases, otherwise the resulting vulnerability would impede us from quantifying and minimizing the actual discrepancy.

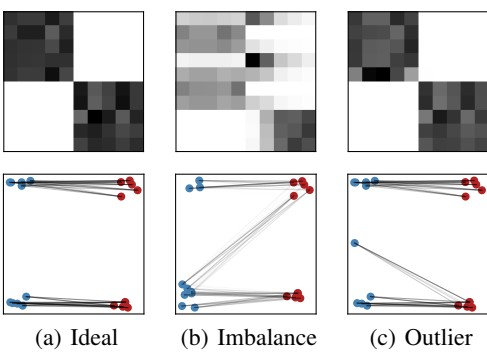

(a) Ideal     (b) Imbalance     (c) Outlier

Figure 2: Optimal transport plan (upper) and its geometric interpretation (down) in three cases, where the connection strength represents the transported mass. Different colors and vertical positions indicate different treatments and outcomes, respectively.

We consider three sampling cases in Figure 2, where the transport strategy is reasonable and applicable in the ideal sampling case in Figure 2(a). However, the transport strategy calculated with (6) is vulnerable to disturbances caused by non-ideal sampling conditions. For instance, Figure 2(b) reveals false matching of units with disparate outcomes in the sampled mini-batch where the outcomes of the two groups are imbalanced; Figure 2(c) showcases false matching of the mini-batch outliers with normal units, causing a substantial disruption of the

transport strategy for other units. Therefore, the vanilla OT in (6) fails to quantify the group discrepancy for producing erroneous transport strategies in non-ideal mini-batches, thereby misleading the update of the representation mapping $\psi$. We term this issue the mini-batch sampling effect (MSE).

Furthermore, this MSE issue is not exclusive to OT-based methods but is also prevalent in other representation-based approaches. Despite this, OT offers a distinct advantage: it allows the formalization of the MSE issue through its mass-preservation constraint, as indicated in (5). This constraint mandates a match for every unit in both groups, regardless of the real-world scenarios, complicating the transport of normal units and the computation of true group discrepancies. This issue is further exacerbated by small batch sizes. Such formalizability through mass-preservation provides a lever for handling the MSE issue, setting OT apart from other representation-based methods [85, 86].

An intuitive approach to mitigate the MSE issue is to relax the marginal constraint, *i.e.*, to allow for the creation and destruction of the mass of each unit. To this end, inspired by unbalanced and weak transport theories [14, 65], a relaxed mass-preserving regularizer (RMPR) is devised in Definition 3.2. The core technical point is to replace the hard marginal constraint in (4) with a soft penalty in (9) for constraining transport strategy. On the basis, the stochastic discrepancy is calculated as

$$\hat{\mathbb{W}}_{\psi}^{\epsilon,\kappa} := \mathbb{W}^{\epsilon,\kappa}\left(\hat{\mathbb{P}}_{\psi}^{\mathrm{T=1}}(r), \hat{\mathbb{P}}_{\psi}^{\mathrm{T=0}}(r)\right), \tag{8}$$

where the hard mass-preservation constraint is removed to mitigate the MSE issue. Building on Lemma 1 in Fatras et al. [23], we derive Corollary 3.1 to investigate the robustness of RMPR to sampling effects, showcasing that the effect of mini-batch outliers is upper bounded by a constant.

**Definition 3.2.** *For empirical distributions $\alpha$ and $\beta$ with $n$ and $m$ units, respectively, optimal transport with relaxed mass-preserving constraint seeks the transport strategy $\pi$ at the minimum cost:*

$$\mathbb{W}^{\epsilon,\kappa}(\alpha,\beta) := \langle \mathbf{D}, \pi \rangle, \pi := \arg\min_{\pi} \langle \mathbf{D}, \pi \rangle - \epsilon \mathrm{H}(\pi) + \kappa(\mathrm{D}_{\mathrm{KL}}(\pi\mathbf{1}_m, \mathbf{a}) + \mathrm{D}_{\mathrm{KL}}(\pi^{\mathrm{T}}\mathbf{1}_n, \mathbf{b})) \tag{9}$$

*where $\mathbf{D} \in \mathbb{R}_{+}^{n \times m}$ is the unit-wise distance, and $\mathbf{a}$, $\mathbf{b}$ indicate the mass of units in $\alpha$ and $\beta$, respectively.*

**Corollary 3.1.** *For empirical distributions $\alpha, \beta$ with $n$ and $m$ units, respectively, adding an outlier $a'$ to $\alpha$ and denoting the disturbed distribution as $\alpha'$, we have*

$$\mathbb{W}^{0,\kappa}(\alpha',\beta) - \mathbb{W}^{0,\kappa}(\alpha,\beta) \le 2\kappa(1 - e^{-\sum_{b\epsilon\beta}(a'-b)^2/2\kappa})/(n+1), \tag{10}$$

*which is upper bounded by $2\kappa/(n+1)$. $\mathbb{W}^{0,\kappa}$ is the unbalanced discrepancy as per Definition 3.2.*

In comparison with alternative methods [9, 83] to relax the marginal constraint, the RMPR implementation in Definition 3.2 enjoys a collection of theoretical properties [65] and can be calculated via the generalized Sinkhorn algorithm [14]. The calculated discrepancy is differentiable *w.r.t.* $\psi$ and thus can be minimized via stochastic gradient methods in an end-to-end manner.

### 3.3 Proximal factual outcome regularizer for unobserved confounders

Existing representation-based methods fail to eliminate the treatment selection bias due to the unobserved confounder effects (UCE). Beginning with CFR [67], the unconfoundedness assumption A.1 (see Appendix A) is often taken to circumvent the UCE issue [75]. In this context, given two units $r_i \in \mathbb{P}_{\psi}^{T=1}(r)$ and $r_j \in \mathbb{P}_{\psi}^{T=0}(r)$, for instance, OT discrepancy in Definition 3.2 calculates the unitwise distance as $\mathbf{D}_{ij} := \|r_i - r_j\|^2$. If Assumption A.1 holds, it eliminates the treatment selection bias since it blocks the backdoor path $X \to T$ in Figure 3(a) by balancing confounders in the latent space. However, Assumption A.1 is usually violated due to the existence of unobserved confounders as per Figure 3(b), which hinders existing methods from handling treatment selection bias since the backdoor path $X' \to T$ is not blocked. The existence of unobserved covariates $X'$ also makes the transport strategy in vanilla OT unidentifiable, which invalidates the calculated discrepancy.

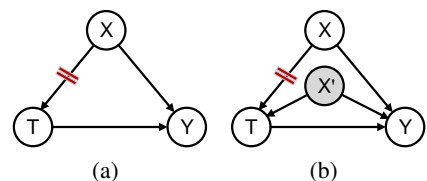

Figure 3: Causal graphs with (a) and w/o (b) the unconfoundedness assumption. The shaded node indicates the hidden confounder $X'$.

To mitigate the effect of $X'$, we introduce a modification to the unit-wise distance. Specifically, we observe that given balanced $X$ and identical $T$, the only variable reflecting the variation of $X'$ is the

outcome $Y$. As such, resonating with Courty et al. [16], we calibrate the unit-wise distance $\mathbf{D}$ with potential outcomes as follow:

$$\mathbf{D}_{ij}^{\gamma} = \|r_i - r_j\|^2 + \gamma \cdot \left[\|y_i^{T=0} - y_j^{T=0}\|^2 + \|y_j^{T=1} - y_i^{T=1}\|^2\right], \tag{11}$$

where $\gamma$ controls the strength of regularization. The underlying regularization is straightforward: units with similar (observed and unobserved) confounders should also have similar potential outcomes. As such, for a pair of units with similar observed covariates, *i.e.*, $\|r_i - r_j\|^2 \approx 0$, if their potential outcomes given the same treatment $t = \{0, 1\}$ differ greatly, *i.e.*, $\|y_i^t - y_j^t\| >> 0$, their unobserved confounders should likewise differ significantly. The vanilla OT technique in (6) where $\mathbf{D}_{ij} = \|r_i - r_j\|^2$ would incorrectly match this pair, generate a false transport strategy, and consequently misguide the update of the representation mapping $\psi$. In contrast, OT based on $\mathbf{D}_{ij}^{\gamma}$ would not match this pair as the difference of unobserved confounders is compensated with that of potential outcomes.

Moving forward, since $y_i^{T=0}$ and $y_j^{T=1}$ in (11) are unavailable due to the missing counterfactual outcomes, the proposed proximal factual outcome regularizer (PFOR) uses their estimates instead. Specifically, let $\hat{y}_i$ and $\hat{y}_j$ be the estimates of $y_i^{T=0}$ and $y_j^{T=1}$, respectively, PFOR refines (11) as

$$\mathbf{D}_{ij}^{\gamma} = \|r_i - r_j\|^2 + \gamma \cdot \left[\|\hat{y}_i - y_j\|^2 + \|\hat{y}_j - y_i\|^2\right], \quad \hat{y}_i = \phi_0(r_i), \quad \hat{y}_j = \phi_1(r_j), \tag{12}$$

Additional justifications, assumptions and limitations of PFOR are discussed in Appendix D.3.

## 3.4 Architecture of entire space counterfactual regression

The architecture of ESCFR is presented in Figure 1(b), where the covariate $X$ is first mapped to the representations $R$ with $\psi(\cdot)$, and then to the potential outcomes with $\phi(\cdot)$. The group discrepancy $\mathbb{W}$ is calculated with the optimal transport equipping with the RMPR in (9) and PFOR in (12).

The learning objective is to minimize the risk of factual outcome estimation and the group discrepancy. Given mini-batch distributions $\hat{\mathbb{P}}^{T=1}(x)$ and $\hat{\mathbb{P}}^{T=0}(x)$ in Definition 3.1, the risk of factual outcome estimation following [68] can be formulated as

$$\mathcal{L}_{\mathrm{F}}(\psi, \phi) := \mathbb{E}_{x_i \in \hat{\mathbb{P}}^{T=1}(x)} \|\phi_1(\psi(x_i)) - y_i\|^2 + \mathbb{E}_{x_j \in \hat{\mathbb{P}}^{T=0}(x)} \|\phi_0(\psi(x_j)) - y_j\|^2, \tag{13}$$

where $y_i$ and $y_j$ are the factual outcomes for the corresponding treatment groups. The discrepancy is:

$$\mathcal{L}_{\mathrm{D}^{\gamma}}^{\epsilon, \kappa}(\psi) := \mathbb{W}^{\epsilon, \kappa, \gamma}\left(\hat{\mathbb{P}}_{\psi}^{\mathrm{T}=1}(r), \hat{\mathbb{P}}_{\psi}^{\mathrm{T}=0}(r)\right), \tag{14}$$

which is in general the optimal transport with RMPR in Definition 3.2, except for the unit-wise distance $\mathbf{D}^{\gamma}$ calculated with the PFOR in (12). Finally, the overall learning objective of ESCFR is

$$\mathcal{L}_{\mathrm{ESCFR}}^{\epsilon, \kappa, \gamma, \lambda} := \mathcal{L}_{\mathrm{F}}(\psi, \phi) + \lambda \cdot \mathcal{L}_{\mathrm{D}^{\gamma}}^{\epsilon, \kappa}(\psi), \tag{15}$$

where $\lambda$ controls the strength of distribution alignment, $\epsilon$ controls the entropic regularization in (5), $\kappa$ controls RMPR in (9), and $\gamma$ controls PFOR in (12). The learning objective (15) mitigates the selection bias following Theorem 3.1 and handles the MSE and UCE issues.

The optimization procedure consists of three steps as summarized in Algorithm 3 (see Appendix B). First, compute $\boldsymbol{\pi}^{\epsilon, \kappa, \gamma}$ by solving the OT problem in Definition 3.2 with Algorithm 2 (see Appendix B), where the unit-wise distance is calculated with $\mathbf{D}^{\gamma}$. Second, compute the discrepancy in (14) as $\langle \boldsymbol{\pi}^{\epsilon, \kappa, \gamma}, \mathbf{D}^{\gamma}\rangle$, which is differentiable to $\psi$ and $\phi$. Finally, calculate the overall loss in (15) and update $\psi$ and $\phi$ with stochastic gradient methods.

# 4 Experiments

## 4.1 Experimental setup

**Datasets.** Missing counterfactuals impede the evaluation of PEHE using observational benchmarks. Therefore, experiments are conducted on two semi-synthetic benchmarks [67, 85], *i.e.*, IHDP and ACIC. Specifically, IHDP is designed to estimate the effect of specialist home visits on infants' potential cognitive scores, with 747 observations and 25 covariates. ACIC comes from the collaborative perinatal project [51], and includes 4802 observations and 58 covariates.

Table 1: Performance (mean±std) on the PEHE and AUUC metrics. "*" marks the baseline estimators that ESCFR outperforms significantly at p-value < 0.05 over paired samples t-test.

| Dataset | ACIC (PEHE) | | IHDP (PEHE) | | ACIC (AUUC) | | IHDP (AUUC) | |
|---|---|---|---|---|---|---|---|---|
| Model | In-sample | Out-sample | In-sample | Out-sample | In-sample | Out-sample | In-sample | Out-sample |
| OLS | 3.749±0.080* | 4.340±0.117* | 3.856±6.018 | 5.674±9.026 | 0.843±0.007 | 0.496±0.017* | 0.652±0.050 | 0.492±0.032* |
| R.Forest | 3.597±0.064* | 3.399±0.165* | 2.635±3.598 | 4.671±9.291 | 0.902±0.016 | 0.702±0.026* | 0.736±0.142 | 0.661±0.259 |
| S.Learner | 3.572±0.269* | 3.636±0.254* | 1.706±1.600* | 3.038±5.319 | **0.905±0.041** | 0.627±0.014* | 0.633±0.183 | 0.702±0.330 |
| T.Learner | 3.429±0.142* | 3.566±0.248* | 1.567±1.136* | 2.730±3.627 | 0.846±0.019 | 0.632±0.020* | 0.651±0.179 | 0.707±0.333 |
| TARNet | 3.236±0.266* | 3.254±0.150* | 0.749±0.291 | 1.788±2.812 | 0.886±0.046 | 0.662±0.014* | 0.654±0.184 | 0.711±0.329 |
| C.Forest | 3.449±0.101* | 3.196±0.177* | 4.018±5.602* | 4.486±8.677 | 0.717±0.005* | 0.709±0.018* | 0.643±0.141 | 0.695±0.294 |
| k-NN | 5.605±0.168* | 5.892±0.138* | 2.208±2.233* | 4.319±7.336 | 0.892±0.007* | 0.507±0.034* | 0.725±0.142 | 0.668±0.299 |
| O.Forest | 8.094±4.669* | 4.148±2.224* | 2.605±2.418* | 3.136±5.642 | 0.744±0.013 | 0.699±0.022* | 0.664±0.157 | 0.702±0.325 |
| PSM | 5.228±0.154* | 5.094±0.301* | 3.219±4.352* | 4.634±8.574 | 0.884±0.010 | 0.745±0.021 | **0.740±0.149** | 0.681±0.253 |
| BNN | 3.345±0.233* | 3.368±0.176* | 0.709±0.330 | 1.806±2.837 | 0.882±0.033 | 0.645±0.013* | 0.654±0.184 | 0.711±0.329 |
| CFR-MMD | 3.182±0.174* | 3.357±0.321* | 0.777±0.327 | 1.791±2.741 | 0.871±0.032 | 0.659±0.017* | 0.655±0.183 | 0.710±0.329 |
| CFR-WASS | 3.128±0.263* | 3.207±0.169* | 0.657±0.673 | 1.704±3.115 | 0.873±0.029 | 0.669±0.018* | 0.656±0.187 | 0.715±0.311 |
| ESCFR | **2.252±0.297** | **2.316±0.613** | **0.502±0.252** | **1.282±2.312** | 0.796±0.030 | **0.754±0.021** | 0.665±0.166 | **0.719±0.311** |

**Baselines.** We consider three groups of baselines. (1) Statistical estimators: least square regression with treatment as covariates (OLS), random forest with treatment as covariates (R.Forest), a single network with treatment as covariates (S.learner [36]), separate neural regressors for each treatment group (T.learner [36]), TARNet [67]; (2) Matching estimators: propensity score match with logistic regression (PSM [60]), k-nearest neighbor (k-NN [18]), causal forest (C.Forest [73]), orthogonal forest (O.Forest [73]); (3) Representation-based estimators: balancing neural network (BNN [31]), counterfactual regression with MMD (CFR-MMD) and Wasserstein discrepancy (CFR-WASS) [67].

**Training protocol.** A fully connected neural network with two 60-dimensional hidden layers is selected to instantiate the representation mapping $\psi$ and the factual outcome mapping $\phi$ for ESCFR and other neural network based baselines. To ensure a fair comparison, all neural models are trained for a maximum of 400 epochs using the Adam optimizer, with the patience of early stopping being 30. The learning rate and weight decay are set to $1e^{-3}$ and $1e^{-4}$, respectively. Other settings of optimizers follow Kingma and Ba [35]. We fine-tune hyperparameters within the range in Figure 5, validate performance every two epochs, and save the optimal model for test.

**Evaluation protocol.** The PEHE in (3) is the primary metric for performance evaluation [67, 85]. However, it is unavailable in the model selection phase due to missing counterfactuals. As such, we use the area under the uplift curve (AUUC) [2] to guide model selection, which evaluates the ranking performance of the CATE estimator and can be computed without counterfactual outcomes. Although AUUC is not commonly used in treatment effect estimation, we report it as an auxiliary metric for reference. The within-sample and out-of-sample results are computed on the training and test set, respectively, following the common settings [44, 45, 67, 85].

## 4.2 Overall performance

Table 1 compares ESCFR and its competitors. Main observations are noted as follows.

- Statistical estimators demonstrate competitive performance on the PEHE metric, with neural estimators outperforming linear and random forest methods due to their superior ability to capture non-linearity. In particular, TARNet, which combines the advantages of T-learner and S-learner, achieves the best overall performance among statistical estimators. However, the circumvention to treatment selection bias results in inferior performance.

- Matching methods such as PSM exhibit strong ranking performance, which explains their popularity in counterfactual ranking practice. However, their relatively poor performance on the PEHE metric limits their applicability in counterfactual estimation applications where accuracy of CATE estimation is prioritized, such as advertising systems.

- Representation-based methods mitigate the treatment selection bias and enhance overall performance. In particular, CFR-WASS reaches an out-of-sample PEHE of 3.207 on ACIC, advancing most statistical methods. However, it utilizes the vanilla Wasserstein discrepancy, wherein the MSE and UCE issues impede it from solving the treatment selection bias. The proposed ESCFR

Table 2: Ablation study (mean±std) on the ACIC benchmark. "*" marks the variants that ESCFR outperforms significantly at p-value < 0.01 over paired samples t-test.

| | | | In-sample | | Out-sample | |
|---|---|---|---|---|---|---|
| SOT | RMPR | PFOR | PEHE | AUUC | PEHE | AUUC |
| ✗ | ✗ | ✗ | 3.2367±0.2666* | **0.8862+0.0462** | 3.2542±0.1505* | 0.6624+0.0149* |
| ✓ | ✗ | ✗ | 3.1284±0.2638* | 0.8734±0.0291 | 3.2073+0.1699* | 0.6698+0.0187* |
| ✓ | ✓ | ✗ | 2.6459+0.2747* | 0.8356±0.0286 | 2.7688±0.4009 | 0.7099+0.0157* |
| ✓ | ✗ | ✓ | 2.5705±0.3403* | 0.8270±0.0341 | 2.6330±0.4672 | 0.7110±0.0287* |
| ✓ | ✓ | ✓ | **2.2520±0.2975** | 0.7968±0.0307 | **2.3165+0.6136** | **0.7542±0.0202** |

achieves significant improvement over most metrics compared with various prevalent baselines[3]. Combined with the comparisons above, we attribute its superiority to the proposed RMPR and PFOR regularizers, which accommodate ESCFR to the situations where MSE and UCE exist.

## 4.3 Ablation study

In this section, to further verify the effectiveness of individual components, an ablation study is conducted on the ACIC benchmark in Table 2. Specifically, ESCFR first augments TARNet with the stochastic optimal transport to align the confounders in the representation space, as described in Section 3.1, which reduces the out-of-sample PEHE from 3.254 to 3.207. Subsequently, it mitigates the MSE issue with RMPR as per Section 3.2 and the UCE issue with PFOR as per Section 3.3, which reduces the out-of-sample PEHE to 2.768 and 2.633, respectively. Finally, ESCFR combines the RMPR and PFOR in a unified framework as detailed in Section 3.4, which further reduces the value of PEHE and advances the best performance of other variants.

## 4.4 Analysis of relaxed mass-preserving regularizer

Most prevalent methods fail to handle the label imbalance and mini-batch outliers in Figure 2(b-c). Figure 4 shows the transport plan generated with RMPR in the same situations, where RMPR alleviates the MSE issue in both bad cases. Initially, RMPR with $\kappa = 50$ presents similar matching scheme with Figure 2, since in this setting the loss of marginal constraint is strong, and the solution is thus similar to that of the vanilla OT problem in Definition 2.3; RMPR with $\kappa = 10$ further looses the marginal constraint and avoids the incorrect matching of units with different outcomes; RMPR with $\kappa = 2$ further gets robust to the outlier's interference and correctly matches the remaining units. This success is attributed to the relaxation of the mass-preserving constraint according to Section 3.2.

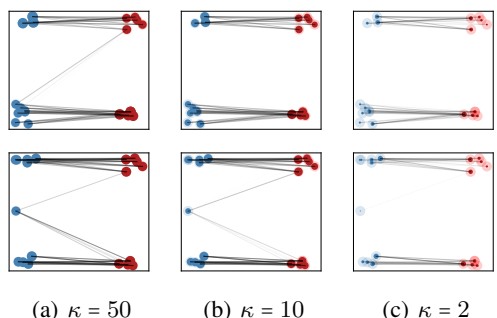

(a) $\kappa = 50$  (b) $\kappa = 10$  (c) $\kappa = 2$

Figure 4: Geometric interpretation of OT plan with RMPR under the outcome imbalance (upper) and outlier (down) settings. The dark area indicates the transported mass of a unit, *i.e.*, marginal of the transport matrix $\pi$. The light area indicates the total mass.

Notably, RMPR does not transport all mass of a unit. The closer the unit is to the overlapping zone in a batch, the greater the mass is transferred. That is, RMPR adaptively matches and pulls closer units that are close to the overlapping region, ignoring outliers, which mitigates the bias of causal inference methods in cases where the positivity assumption does not strictly hold. Current approaches mainly achieve it by manually cleaning the data or dynamically weighting the units [32], while RMPR naturally implements it via the soft penalty in (9).

We further investigate the performance of RMPR under different batch sizes and $\kappa$ in Appendix D.2 to verify the effectiveness of RMPR more thoroughly.

---

[3]An exception would be the within-sample AUUC, which is reported over training data and thus easy to be overfitted. This metric is not critical as the factual outcomes are typically unavailable in the inference phase. We mainly rely on out-of-sample AUUC instead to evaluate the ranking performance and perform model selection.

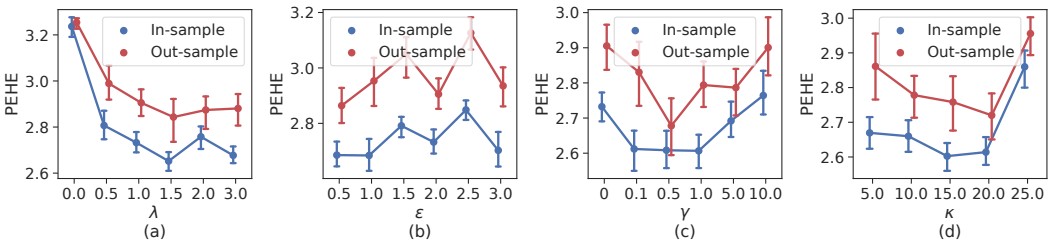

Figure 5: Parameter sensitivity of ESCFR, where the lines and error bars indicate the mean values and 90% confidence intervals, respectively. (a) Impact of alignment strength ($\lambda$). (b) Impact of entropic regularization strength $\epsilon$. (c) Impact of PFOR strength $\gamma$ ($\times 10^3$). (d) Impact of RMPR strength $\kappa$.

## 4.5 Parameter sensitivity study

In this section, we investigate the role of four critical hyperparameters within the ESCFR framework—namely, $\lambda$, $\epsilon$, $\kappa$, and $\gamma$. These hyperparameters have a profound impact on learning objectives and the performance, as substantiated by the experimental results presented in Figure 5.

First, we vary $\lambda$ to investigate the efficacy of stochastic optimal transport. Our findings suggest that a gradual increase in the value of $\lambda$ leads to consistent improvements. For instance, the out-of-sample PEHE diminishes from 3.22 at $\lambda = 0$ to roughly 2.85 at $\lambda = 1.5$. Nonetheless, overly emphasizing distribution balancing within a multi-task learning framework can result in compromised factual outcome estimation and, consequently, suboptimal CATE estimates. The next parameter scrutinized is $\epsilon$, which serves as an entropic regularizer. Our study shows that a larger $\epsilon$ accelerates the computation of optimal transport discrepancy [24]. However, this computational advantage comes at the cost of a skewed transport plan, as evidenced by an increase in out-of-sample PEHE with oscillations.

Following this, we explore the hyperparameters $\gamma$ and $\kappa$, associated with PFOR and RMPR, respectively. Extending upon vanilla optimal transport, we incorporate unit-wise distance via (12) and assess the impact of PFOR. Generally, the inclusion of PFOR positively influences CATE estimation. However, allocating excessive weight to proximal outcome distance compromises the performance, as it dilutes the role of the unit-wise distance in the representation space. Concurrently, we modify the transport problem with (9) to examine the implications of RMPR. Relieving the mass-preserving constraints through RMPR significantly enhances CATE estimation, but an excessively low value of $\kappa$ hampers performance as it fails to ensure that the representations across treatment groups are pulled closer together in the optimal transport paradigm.

## 5 Related works

Current research aims to alleviate treatment selection bias through the balancing of distributions between treated and untreated groups. These balancing techniques can be broadly categorized into three categories: reweighting-based, matching-based, and representation-based methods.

Reweighting-based methods mainly employ propensity scores to achieve global balance between groups. The core procedure comprises two steps: the estimation of propensity scores and the construction of unbiased estimators. Propensity scores are commonly estimated using logistic regression [11, 20, 37, 88]. To enhance the precision of these estimates, techniques such as feature selection [69, 76, 77], joint optimization [88, 89], and alternative training techniques [92] have been adopted. The unbiased estimator is exemplified by the inverse propensity score method [59], which inversely re-weights each units with the estimated propensity scores. While theoretically unbiased, it suffers from high variance with low propensity and bias with incorrect propensity estimates [39, 40, 74]. To alleviate these drawbacks, doubly robust estimators and variance reduction techniques have been introduced [26, 42, 58]. Nevertheless, these methods remain constrained by their reliance on propensity scores, affecting their efficacy in real-world applications.

Matching-based methods aim to match comparable units from different groups to construct locally balanced distributions. The key distinctions between representative techniques [8, 46, 60, 79] lie in their similarity measures. A notable exemplar is the propensity score matching approach [60], which

computes unit (dis)similarity based on estimated propensity scores. Notably, Tree-based methods [73] can also be categorized as matching approaches, but use adaptive similarity measures. However, these techniques are computationally intensive, limiting their deployment in large-scale operations.

Representation-based methods aim to construct a mapping to a feature space where distributional discrepancies are minimized. The central challenge lies in the accurate calculation of such discrepancies. Initial investigations focused on maximum mean discrepancy and vanilla Wasserstein discrepancy [31, 67], later supplemented by local similarity preservation [85, 86], feature selection [13, 27], representation decomposition [27, 80] and adversarial training [87] mechanisms. Despite their success, they fail under specific but prevalent conditions, such as outlier fluctuations [23] and unlabeled confounders [90], undermining the reliability of calculated discrepancies.

The recent exploration of optimal transport in causality [71] has spawned innovative reweighting-based [21], matching-based [10] and representation-based methods [44, 45]. For example, Li et al. [45] utilize OT to align factual and counterfactual distributions; Li et al. [44] and Ma et al. [48] use OT to reduce confounding bias. Despite these advancements, they largely adhere to the vanilla Kantorovich problem that corresponds to the canonical Wasserstein discrepancy, akin to Shalit et al. [67]. Adapting OT problems to meet the unique needs of treatment effect estimation remains an open area for further research.

## 6 Conclusion

Due to the effectiveness of mitigating treatment selection bias, representation learning has been the primary approach to estimating individual treatment effect. However, existing methods neglect the mini-batch sampling effects and unobserved confounders, which hinders them from handling the treatment selection bias. A principled approach named ESCFR is devised based on a generalized OT problem. Extensive experiments demonstrate that ESCFR can mitigate MSE and UCE issues, and achieve better performance compared with prevalent baseline models.

Looking ahead, two research avenues hold promise for further exploration. The first avenue explores the use of normalizing flows for representation mapping, which offers the benefit of invertibility [12] and thus aligns with the assumptions set forth by Shalit et al. [67]. The second avenue aims to apply our methodology to real-world industrial applications, such as bias mitigation in recommendation systems [74], an area currently dominated by high-variance reweighting methods.

## Acknowledgements

This work is supported by National Key R&D Program of China (Grant No. 2021YFC2101100), National Natural Science Foundation of China (62073288, 12075212, 12105246, 11975207) and Zhejiang University NGICS Platform.

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
