# OpenReview forum: "Optimal Transport for Treatment Effect Estimation"
_NeurIPS.cc/2023/Conference — NeurIPS 2023 poster_

### Official Review · Reviewer_jCP6 · 2023-06-16

**Soundness:** 2 fair
**Presentation:** 3 good
**Contribution:** 2 fair
**Rating:** 7
**Confidence:** 4

**Summary:**

In this paper, authors have applied optimal transport to address selection-bias issue in binary treatment setting for individualized treatment effect estimation. They have proposed a relaxed mass-preserving regularizer to address the mis-alignment generated from outcome imbalance and outliers in non-ideal mini-batches. They also proposed a proximal factual outcome regularizer to relax the unconfoundedness assumption. They have done extensive evaluation of the proposed approach on IHDP and ACIC, two semi-synthetic datasets.

**Strengths:**

- A novel method is proposed for treatment effect estimation in binary setting based on optimal transport (OT) that uses a relaxed mass-preserving regularizer to address the mis-alignment generated from outcome imbalance and outliers in non-ideal mini-batches, as well as a proximal factual outcome regularizer to relax the unconfoundedness assumption.
- Extensive evaluations is performed to study the proposed model that provides necessary analysis and ablation studies.
- Theoretical results are provided to bound the error in terms of the proposed components to show the convergence (I have not verified the proof).
- Idea was clearly stated with sufficient details.


**Weaknesses:**

- Baselines used in the paper seem old. There are very large number of methods proposed for treatment effect (TE) estimation. It would have been nice to see comparison with recent techniques. Hoever, even if authors don't add new baselines, I think paper has sufficient novelty to be above the acceptance threshold.
- The paper solves (equation (1)) Conditional average treatment effect estimation also called as heterogeous treatment effects estimation, as called in machine learning literature, but paper did not even mention these names. Please update the paper so that readers don't have any confusion. There is already a lot of confusion in ML regarding the terminology.
- Causal inference methods are typically based on some assumptions -- but I did not see any. Please clearly state the assumptions required to apply these methods.

**Questions:**

- Training all networks for 400 epochs do not look fair to me. Are all the methods getting trained before 400 epochs or not? In my experience this affects the performance in causal inference. All the networks should be trained using early stopping and by setting large number of max-epochs so that every learner gets chance to train properly.
- Have you used two layers each in representation learning and outcome mapping or total two layers for both?
- First line of Introduction section does not look right to me. Doesn't RCT mostly answer population-level or heterogeneous-level questions but not the individual-level treatment effects?
- In figures 1(a) and 2, please clearly state what are the different axis, e.g., for Figure 2, it is not clear what is x-axis or what is y-axis?
- Why does title says 'a new perspective' when the idea is very widely used, including in causal inference, as authors have themselves discussed that?

**Limitations:**

Authors did not discuss the assumptions and limitations of thier work.

---

> ### Author Rebuttal · Authors · 2023-08-09
>
> **[W1] Comparison with recent techniques**
> - Thanks for your kind recognition.
> We agree that comparison with recent techniques could improve this work. Thus, we incorporated two more recently baselines founded on representation learning, DRCFR [1] and DRCFR-MIM [2], in appendix, with the aim of modernizing our scope of baselines.
> These two methods introduce variable decomposition and employ the distribution alignment technology (CFR) as a component.
> Therefore, *it would be somewhat unfair for ESCFR to be directly compared to them, given that ESCFR is not equipped with variable decomposition technology*. As a result, we elected to place these results in appendix rather than the main content of the paper to ensure a fair presentation while preserving completeness.
>
> - Here, we provide the comparisons (Table 6 in the appendix). DRCFR and DRCFR-MIM exhibited more stable performance (with less variance) and did surpass CFR-WASS in terms of out-of-sample PEHE. Nonetheless, ESCFR significantly outperforms them across most metrics.
>
>   | Dataset   | Out-sample    | Out-sample    | In-sample     | In-sample     |
>   |-----------|---------------|---------------|---------------|---------------|
>   | Model     | PEHE          | AUUC          | PEHE          | AUUC          |
>   | TARNet    | 3.2054±0.2676 | 0.6683±0.0139 | 3.2933±0.4076 | 0.8597±0.0350 |
>   | CFR-WASS  | 3.0819±0.3407 | 0.6693±0.0250 | 3.4280±0.2857 | 0.8439±0.0349 |
>   | DRCFR     | 3.0199±0.0658 | 0.6360±0.0080 | 3.4957±0.0895 | 0.8456±0.0073 |
>   | DRCFR-MIM | 2.9593±0.1532 | 0.6456±0.0189 | 3.3875±0.2208 | 0.8434±0.0093 |
>   | ESCFR     | 2.3082±0.4334 | 0.7311±0.0123 | 2.5385±0.5020 | 0.8416±0.0367 |
>
> [1] Negar Hassanpour, et al. Learning disentangled representations for counterfactual regression. In ICLR. 2020. \
> [2] Mingyuan Cheng, et al. Learning disentangled representations for counterfactual regression via mutual information minimization. In SIGIR. 2022.
>
> **[W2] Terminologies: ITE, CATE and HTE**
>
> - We sincerely apologize for any confusion caused. Your observation regarding the naming convention is entirely valid, and we appreciate your suggestion. In this paper, **we delve into the individual-level causal effect estimation, where each individual is identified by its covariates $x_i$**, in line with existing works [1,2,3]. Under this circumstance, individual treatment effect estimation (concerning the effect of an individual unit with covariates $x_i$) is interchangeably referred to as CATE or HTE, as you correctly pointed out, in the machine learning literature [1,2,3].
> - We acknowledge the importance of maintaining clarity and consistency in terminology, especially given the existing confusion in the field. Consequently, we commit to updating the paper to employ the term "CATE" and will elucidate the connections between these terminologies in Appendix A.
>
> [3]  Uri Shalit, et al. Estimating individual treatment effect: generalization bounds and algorithms. In ICML. 2017.
>
> **[W3 & Limiations] Assumptions & limitations**
> - Thanks for your suggestion. To maintain the concise presentation of the main paper, we have indeed delineated the necessary assumptions in **Appendix A.1**.
> Additionally, the assumptions and limitations specific to our work are thoroughly discussed in **Appendix D.3**.
>
> **[Q1] Training epochs & early stopping**
> -  We sincerely apologize for an oversight during the proofreading stage, where we inadvertently deleted a critical technique regarding early stopping. Allow us to clarify the following points:
> - The maximum number of training epochs is 400, with early stopping implemented (tolerance=30).
> - We utilized AUUC for model selection and early stopping, as it better aligns with our objectives.
>
> **[Q2] About network structures**
> - We employed a structure consisting of 3 layers for both representation learning and outcome mapping, inspired by CFR. As articulated in CFR: `Both versions are implemented as feed-forward neural networks with 3 fully-connected exponential-linear layers for the representation and 3 for the hypothesis.`
> - In terms of key parameters, such as the number of hidden nodes and layers, we fine-tuned them based on TARNet since most representation-based methods can be viewed as a regularization on TARNet. Moreover, we ensured that all deep models possessed either the same (at least similar) number of parameters to facilitate a fair comparison.
>
> **[Q3] About the first line of Introduction**
> - We sincerely apologize for the confusion caused. In this paper, we study the individual-level causal effect estimation, **where each individual is identified by its covariates $x_i$** as [1,2,3], which is also known as CATE and HTE (see our response to W1). RCT can successfully avoid the treatment selection bias issue and the unobserved confounding issue since it cuts the edge between the covariates and treatment in the causal graph. Since in RCT, there is not selection bias, vast machine learning models (e.g., XGBoost, T-learner) can be used without adjustment, which therefore helps ITE estimation.
>
> **[Q4] Fig. 1(a) and 2: meanings of x- and y-axis**
> - Sorry for the confusion caused. In both figure 1(a) and 2, the x-axis represents the covariates, and the y-axis represents the outcomes.
>
> **[Q5] 'a new perspective' in title**
> - We note that although there are some attempts that use OT to empower causal inference, these attempts are fairly few and remain in an initial exploratory stage. More importantly, there has not been attempt to improve current **representation-based estimators** by devising OT algorithm, with the aim at mitigating the specific issues in causal inference. We hope to regard this paper as a exemplar to show the potential and flexibility of OT in this field, which we believe offers a fresh perspective for representation-based solutions. We are also pleased to receive any further comments or suggestions you may have in this regard. We look forward to hearing from you.

---

> > ### Comment · Reviewer_jCP6 · 2023-08-10
> >
> > Thank you for your responses.
> >
> > Authors' responses have mostly clarified  my comments. However, I am not very convinced with 'a new perspective' in title. From contributions' perspective, yes it is new take to solve CATE problem but not from OT perspective. After reading the title, readers will think it in terms of OT and not in terms of issues addressed in the paper. Although, this should not be a show stropper.
> >
> > I have quickly skimmed through comments of other reviewers. One major comment was about 'incremental novelty'. I think, here what issue is solved is more important than how it is solved. Since the paper addresses important issues of mini-batch sampling effects (MSE) and unobserved confounder effects (UCE) which were overlooked earlier so the paper has sufficient novelty. However, reviewer MG3J has raised a valid point regarding UCE and its evaluation. I would be keen to follow that discussion.
> >
> > Based on above points, I raise my score from '6: Weak Accept' to '7: Accept'.

---

> > > ### Author Response · Authors · 2023-08-14
> > > **Thank you for your follow-up!**
> > >
> > > We sincerely appreciate your prompt response, recognition of our work, and decision after reviewing feedback from other reviewers. Your suggestion will be thoroughly taken into account, and we'll discuss the choice of the title carefully during the revision process. We warmly welcome any inquiries you may have regarding our work and will make every effort to answer them during the discussion phase.

---

### Official Review · Reviewer_MG3J · 2023-06-18

**Soundness:** 2 fair
**Presentation:** 4 excellent
**Contribution:** 3 good
**Rating:** 6
**Confidence:** 4

**Summary:**

The authors propose to minimize a (relaxed) Sinkhorn distance between representations under the two treatment groups in order to balance the latent space and better predict the treatment effects. They highlight two problems with existing TARNet-based approaches that do not regularize with optimal-transport cost: mini-batch sampling effects (MSE) and unobserved confounder effects (UCE). Theorem 3.1 & Corollary 3.1 formally characterize how the proposed method suffers less from MSE. The authors also introduce additional regularization terms that are meant to deal with UCE. Finally, they show improved performance with two classical semi-synthetic benchmarks.

**Strengths:**

 * The problem of estimating heterogeneous treatment effects is important, the paper is written well, and the solution is creative.
 * The motivation for the relaxed Sinkhorn distance (Sec. 3.2) is great.
 * The benchmark results are solid.
 * The additional empirical explorations (Sec. 4.3 - 4.5) are interesting as well.

**Weaknesses:**

My biggest problem is with the UCE part of the paper.
 * Sec. 3.3: the explanation for the hidden-confounding regularizer makes intuitive sense. However, I do not see a formal connection. Once the authors substitute the actual potential outcomes with the model's predictions (Eq. 11 -> Eq. 12), one begins to wonder if this can actually help with hidden confounding.
 * A major confounder with the issue above is that the authors do not appear to include hidden confounders in their semi-synthetic benchmarks. Therefore, it is impossible to tell if this regularization approach really helps with UCE.
 * If the authors still wish to make the case for UCE mitigation, they need to reference the growing literature on partial identification / sensitivity analysis for hidden confounders and how their regularization relates to sensitivity models.

Besides that, I think the authors should spend more time addressing what the theoretical results mean for the MSE issue that they highlighted in the introduction.

Small nitpicks:
* Figure 1a, the orange annotation looks like it is referring to the X_1 part of the figure.
 * line 107-108: "Optimal transport is a preferred method due to its advantages over competitors." is overly vague.
 * line 108-109, "It accounts for the distribution’s geometry, making it effective [where the KL-divergence] fails." is a strange statement. Why does the geometry help here? No explanation is provided.
 * The beginning of the paper is a bit verbose.

**Questions:**

How do you know that you address UCE? As the paper stands, I recommend that you refrain from discussing hidden confounders in your motivation for the method and stick with what really *can* be shown for your proposed regularization. I am willing to raise my score once this is addressed.

Is MSE really an issue with previous approaches? Can you show that clearly?

**Limitations:**

Limitations or potential societal impacts are not discussed.

---

> ### Author Rebuttal · Authors · 2023-08-09
>
> **[W1] About the hidden-confounding regularizer**
> - **Formal connection.** Please refer to our response to the Reviewer VwsR, section "Theoretical Guarantees for the UCE Regularizer".
> - **Substituting Potential Outcomes with Model Predictions.** Pseudo label (using model predictions to substitue ground-truth labels) has been shown effective in both causal inference and machine learning areas.
>   - In causal inference, doubly robust learning uses an imputation model to impute labels or errors for unobserved data (without ground-truth labels), and use the imputed values as label for model training to achieve double robustness. It is wodely used in treatment effect estimation [1] and debiasing recommendation [2].
>   - In machine learning, for example, in weakly supervised learning, wrapper methods (e.g., self-training), as one of the oldest and prevalent methods, augment the labeled training data with unlabeled data whose labels are obtained from model predictions [6]. In robust machine learning, it is shown that pseudo label magnify useful underlying information in data, and thus can be used to correct noisy labels [3-5].
>   - Thus, using model prediction is a reasonable practice, which has also been shown effective empirically.
>
> [1] Estimation of regression coefficients when some regressors are not always observed. Journal of the American statistical Association. 1994. \
> [2] Doubly Robust Joint Learning for Recommendation on Data Missing Not at Random. ICML. 2019. \
> [3] Who said what: Modeling individual labelers improves classification. AAAI. 2018. \
> [4] Noise against noise: stochastic label noise helps combat inherent label noise. ICLR. 2020. \
> [5] A closer look at memorization in deep networks. ICML. 2017. \
> [6] A survey on semi-supervised learning. Machine learning. 2020.
>
> **[W2] About experiments on semi-synthetic benchmarks**
>
> - We do realize this problem. Unobserved confounders are generally only present in real-world observational data. But in observational data, the performance of ITE estimators cannot be evaluated due to missing counterfactuals. As such, we followed the general settings and conduct experiments on semi-synthetic data. **For semi-synthetic data, one source of hidden confounders is the information loss from the raw data space to the representation space, where not all valuable information (e.g., confounders) is extracted and preserved in the representation space.** In this context, the validity of PFOR was verified by ablation and sensitivity studies in Section 4.3 and 4.5.
> - **We note PFOR also works without unobserved confounders.** One source of performance gain would be the statistical regularization, encouraging samples with similar outcomes to share similar representations, which is an effective prior according to nearest-neighbor approaches. Another benefit comes from the prompts that the outcomes provide to the alignment module. Specifically, vanilla Sinkhorn considers each variable equally during the matching phase; in contrast, PFOR highlights variables with large correlation with Y (i.e., more significant confounding effects), which is an effective prompt.
>
> **[W3] About connections with partial identification**
>
> It is indeed an invaluable comment. Please refer to our response to the Reviewer VwsR, which is highlighted as **"Inspiration from Partial Identification"**.
>
> **[Q1] About the regularizer for UCE & the motivations**
>
> There are other alternative explanations for the effectiveness of the proposed PFOR regularizer. To list a few:
> - The performance gain could come from the prompts given to the alignment module. Vanilla Sinkhorn considers each variable equally; in contrast, PFOR focuses on variables with great correlation with Y (i.e., significant confounding effects), providing prompts to the alignment algorithm.
> - The gain could come from the statistical regularization: samples with similar outcomes to share similar representations, which is an effective prior according to nearest-neighbor approaches.
>
> Nevertheless, in this work, we suggest that PFOR help alleviate the UCE issue in very limited cases given the Monotonicity Assumption in Appendix D, instead of functioning as a general solution for addressing it. Following your kind advice, we will refine our narrative in this regard and discuss the possible alternative explanations above in the main text.
>
> **[Q2] About MSE issue with previous approaches**
> - Yes, MSE is a concern in previous representation-based approaches.
>   - **Theoretically**, the complexity term $\mathcal{O}(\cdot)$ in Theorem 3.1 unveils the risks induced by stochastic sampling, which exists in other representation-based methods as well. Precisely, $\mathcal{O}(\cdot)$ emerges from the discrepancy between the entire population and the sampled mini-batch units (refer to Eq.(30)), and is highly susceptible to the uncontrollable quality of sampling. Therefore, **the discrepancy should be robust to bad sampling cases; otherwise, the variance can hinder the computation severely.** This vulnerability is ignored by existing representation-based solutions.
>   - **Empirically**, Figure 2 illustrates that Sinkhorn discrepancy **fails to quantify the group discrepancy for producing erroneous transport strategies** in non-ideal mini-batches, and thus **misguiding the update of the representation mapping $\psi$**. Established methods are notably affected by this issue. We have opened a new thread to discuss it with additional experiments in Appendix D2.
> - **Beyond OT**, most alignment methods, e.g., MMD and adversarial training, grapple with the MSE issue. When faced with a mini-batch with imbalanced outcomes, for example, they naively minimize the discrepancy, erasing discriminative characteristics essential for outcome estimation.
> - **Advantage of OT**, OT allows to **formalize the MSE issue** as the mass-preserving constraint, thereby giving a natural solution: relaxing the marginal constraint, which is the primary intuition behind RMPR.

---

> > ### Comment · Reviewer_MG3J · 2023-08-17
> > **Raising my score to 6**
> >
> > Thank you for your response. After reading your rebuttals as well as Section D.3 more carefully, I think I understand the corresponding sensitivity assumption for hidden confounders that lies behind your regularization approach. I trust that you will clarify this in the main text.
> >
> > You also addressed some of my concerns surrounding the benchmarks. Still, there are many ways to induce hidden confounding in semi-synthetic benchmark, for instance by hiding some of the covariates. This could have been a valuable addition to the study.
> >
> > Overall, I have updated my score to reflect my new assessment of this work.

---

### Official Review · Reviewer_hjjj · 2023-06-26

**Soundness:** 3 good
**Presentation:** 3 good
**Contribution:** 3 good
**Rating:** 5
**Confidence:** 4

**Summary:**

The objective of this research paper is to tackle two primary concerns: (1) the effects of mini-batch sampling (MSE), which lead to misalignment in non-ideal mini-batches characterized by outcome imbalance and outliers, and (2) the impacts of unobserved confounders (UCE), resulting in inaccurate discrepancy calculation due to the neglect of these unobserved factors. To address these challenges, this paper presents a novel approach based on optimal transport in the context of causality.

More specifically, the proposed approach builds upon the stochastic optimal transport framework. It introduces a relaxed mass preserving regularizer to mitigate the MSE issue and devises a proximal factual outcome regularizer to address the UCE problem. By incorporating these techniques, the proposed method achieves a principled solution for treatment selection bias.

The effectiveness of the proposed method is extensively evaluated through a series of experiments. The results demonstrate that the approach successfully overcomes treatment selection bias and outperforms existing state-of-the-art methods by a significant margin.

**Strengths:**

The paper is clearly written and well-presented. It delves into the important question of improving the estimation of individualized treatment effects, a crucial topic within the field of causal inference. Optimal transport, an emerging field in machine learning, is effectively employed to measure treatment selection bias. Furthermore, the two proposed regularizers are technically intriguing and efficient, as supported by both theoretical and empirical evidence.

**Weaknesses:**

I have two main concerns:

1. The potential applicability of the proposed relaxed mass-preserving regularizer extends beyond causal effect estimation and encompasses a wide range of application settings. It appears to be independent of the estimation of causal effects. The authors acknowledge that the OT discrepancy can be easily affected by various sampling cases, highlighting that it is not solely limited to the distribution discrepancy across treatment groups. Consequently, the main contribution of this regularizer may be seen as an incremental addition to the field of causal inference, raising concerns about the originality and novelty of the paper.

2. The existing literature on the subject appears to be considerably understudied. For example, important references like "Optimal Transport for Counterfactual Estimation: A Method for Causal Inference" by Arthur Charpentier, Emmanuel Flachaire, and Ewen Gallic are missing and remain unaddressed in the related work section. It would be beneficial for the authors to conduct a more comprehensive investigation of the related literature and consider incorporating these references into their study.

**Questions:**

See above

**Limitations:**

See above

---

> ### Author Rebuttal · Authors · 2023-08-09
>
> **[W1] The main contribution of the proposed relaxed mass-preserving regularizer may be seen as an incremental addition to the field of causal inference, raising concerns about the originality and novelty of the paper.**
>
> - Thank you for your sincere comments. We acknowledge your concern regarding the contribution of the proposed relaxed mass-preserving regularizer to the field of treatment effect estimation (TEE). However, it is imperative to underline the context and specific challenges our work seeks to address.
> Although the MSE issue is not unique to TEE, this issue takes on particular significance in this field.
> It is specified as two issues, i.e., outcome imbalance and outliers, which directly impacts the reliability of the alignment strategy and thus the causal estimators.
>
> - Notably, this issue has been somewhat overlooked in current progress, while it is present across numerous representation-based methods.
> Our regularizer's value lies in addressing this vital yet neglected issue, specifically within the realm of current TEE strategies based on representation alignment. By relaxing the mass-preserving constraint of the vanilla OT, this challenging issue is mitigated, which improves the TEE performance.
>
> - We also wish to slightly stress that the application of optimal transport within causal inference remains a nascent area of study. Our proposed method represents an interesting attempt and trial in this direction, reflecting a thoughtful response to existing problems and a promising pathway for future exploration.
>
> **[W2] The existing literature on the subject appears to be considerably understudied. For example, important references like [1] are missing and remain unaddressed in the related work section. It would be beneficial to conduct a more comprehensive literature review and consider incorporating these references into their study.**
>
> - Thank you very much for your kind suggestion. The related work section in the main paper seems not comprehensive enough due to the limited page quota. We have included a more comprehensive literature review in appendix. According to the methods used for balancing, current solutions can be divided into matching based methods, re-weighting based methods and distribution alignment methods. We focus on the re-weighting based methods and distribution alignment methods in the related works.
> - We agree that it is essential to comprehensively review literature in the intersection of OT and causal inference, so we have opened a new thread in the related work to give a brief introduciton, and explained our difference with them.
> - Thanks for recommending an important reference.
>     - summary of this paper. Overall, this work first match each unit in the control group with another one in the treat group, and then constructs CATE estimators based on it. The core of this work, in our view, is a matching-based CATE estimator, i.e., the mutatis mutandis uantile-based CATE in Definition 3.1-3.2. The authors further specify the optimal matching strategy as an OT problem in Section 4.
>     - Similarity and dissimilarity. Both works aim to achieve CATE estimation through optimal transport. They both acknowledge the selection bias issue and use OT to construct (locally) balanced representations. However, [1] is a matching-based method, which aims to match similar units in the data space, while this work is a representation-based method, which aims to minimize the distribution discrepancy in the latent space. On top of that, this work mainly cares about the flexibility of OT variants to tackle issues, i.e., MSE and UCE, which are critical in the causal inference field. These regularizations, in our view, could inspire further research based on [1], to refine matching strategies to make it more adaptable to causal inference.
>
> - After careful reading and discussion, we appreciated the insights and contributions of this work, and decided to include it in our related work section.
>
> [1] Charpentier, Arthur et al. "Optimal Transport for Counterfactual Estimation: A Method for Causal Inference." (2023).

---

> > ### Comment · Reviewer_hjjj · 2023-08-16
> > **Response to Authors**
> >
> > I appreciate the detailed response to my questions from the authors. Given that both my evaluation and the author's rebuttal are in accord about the significance and potential of the crossover between OT and causal inference as a field, as well as the concerns surrounding the proposed method's contribution, I believe that my rating can adequately convey my standpoint on this paper. As a result, I intend to keep my score.

---

### Official Review · Reviewer_VwsR · 2023-07-05

**Soundness:** 4 excellent
**Presentation:** 4 excellent
**Contribution:** 3 good
**Rating:** 8
**Confidence:** 5

**Summary:**

The paper introduces a new approach called Entire Space CounterFactual Regression (ESCFR) to estimate individual treatment effects from observational data, addressing two critical problems that existing methods fail to solve. The first problem is mini-batch sampling effects (MSE), which lead to misalignment in non-ideal mini-batches with outcome imbalance and outliers. The second problem is unobserved confounder effects (UCE), which result in inaccurate discrepancy calculation due to the neglect of unobserved confounders. ESCFR combines the principles of optimal transport and causality to overcome these issues.

**Strengths:**

- The theoretical evidence is solid and convincing.
- The topic, mentioned in the paper as the mini-batch misalignment and UCE error, are interesting and practical issues of ITE estimation.
- Experimental results are solid.

**Weaknesses:**

Is there some theoretical guarantee for the UCE regularizer proposed in this paper? If so, I think the quality can be further improved.

**Questions:**

See in Weaknesses

**Limitations:**

See in Weaknesses

---

> ### Author Rebuttal · Authors · 2023-08-08
>
>
> We extend our sincere gratitude for your insightful comments and constructive critique. We acknowledge your point concerning the theoretical support of the UCE regularizer. We are pleased to elaborate our thoughts and methodologies to clarify the theoretical foundation.
>
> **Is there some theoretical guarantee for the UCE regularizer proposed in this paper? If so, I think the quality can be further improved.**
>
> - **Theoretical Guarantees for the UCE Regularizer.** The UCE regularizer's main theoretical backing derives from the **flexibility of optimal transport in defining distribution geometry** within latent space.
>
>   - **Problem Definition:** The core issue revolves around measuring distribution discrepancies between treated and controlled groups in the presence of unobserved confounder effects. By estimating this discrepancy and minimizing it as a loss function, we encourage the overlapping of units from different groups in latent space, thereby controlling the generalization risks of outcome mappings $\phi$ within both treatment groups.
>
>   - **Role of OT:** OT enables us to **calculate discrepancy with unit-wise distance.** Not only does this facilitate computation, but more importantly, it enables the **editing of geometry according to task characteristics** in the latent space. Specifically, according to the Kantorovich problem, we have $\mathbb{W}(\alpha, \beta):=\min_{\boldsymbol{\pi} \in \Pi(\alpha, \beta)}\langle\mathbf{D}, \boldsymbol{\pi}\rangle$. Then, we edit the geometry of latent space via unit-wise distance $\mathbf{D}$. In general, we can use the Euclidean geometry, where the distance between the i-th treated unit and the j-th untreated unit is $\mathbf{D}_{ij}=||x_i-x_j||_2^2$ , with $x$ denoting observed confounders. To consider unobserved confounders $x^\prime$ , we can define the unit-wise distance as $||x_i-x_j||_2^2+||x_i^{\prime}-x_j^{\prime}||_2^2$ , which is so far grounded in rigorous theoretical support.
>    - **Outcome Calibration:** Given the monotonic assumption, we calibrate the distance with the outcome difference rather than unobserved covariates. Since $x^\prime$ is not observable, a natural idea is that given similar observed covariates, i.e., $x_i=x_j$ and treatment indicator, the effect of unobserved confounder should be reflected on the outcomes given the same treatment. As such, it is reasonable to use the outcome difference to calibrate $\mathbf{D}_{ij}$, given the assumption that there exists a positive relationship between unobserved confounders and potential outcomes. We discussed the assumption as a limitation of this work in Appendix D.3.
>
> - **Inspiration from Partial Identification.**
>   - **Partial identification:** An estimand $\theta$ is partially identified if the observed data distribuion is compatible with multiple values of $\theta$.  In causal inference, challenges like unobserved confoundings might prevent precise causal effect pinpointing. A weaker alternative is to obtain a range of possible causal effects, known as a "identified set", and reduce the size of the set using proper assumptions.  For example, given assumptions, e.g., monotone treatment selection, we can narrow the bound of treatment effect. More recently, according to Lu and Ding's formulation [1], assuming that we know two sensitivity parameters $\frac{\mathbb{E}(Y(1) \mid Z=1, X)}{\mathbb{E}(Y(1) \mid Z=0, X)}=\varepsilon_1(X)$, $\frac{\mathbb{E}(Y(0) \mid Z=1, X)}{\mathbb{E}(Y(0) \mid Z=0, X)}=\varepsilon_0(X)$, we can identify the treatment effect with unmeasured confounding.
>
>   - **Connection with partial identification:** There are significant similarities between the proposed PFOR regularizer and partial identification principles. One significant similarity is that the transport strategy, i.e., $\pi$ in Eq.(2), is partially identified given the existence of unobserved confounders. More specifically, assuming that the unobserved confounder $x^\prime$ has multiple candidate values, there should be multiple corresponding transport strategies, so we can only identify a stratege set. Then, making the monotonic assumption between $x^\prime$ and outcomes, we calibrate the unit-wise distance with the outcome differences, to reduce the size of strategy set and achieve more accurate estimation among possible transport strategies.
>   - **Assumption Criticality:** In both works, whether the assumptions made fit the application is a crucial aspect. In our work, we assume that the unobserved confounders are monotone with the outcome given $x$ and $t$, as is formulated in Assumption D1. We note that this assumption is also employed in emerging partial identification work [2], and its plausibility is analyzed in Appendix D.
>
> - **Conclusion and Future Directions.**
> We recognize that our method only mitigates unobserved confounding effects in specific cases where the assumptions are met. Nevertheless, we maintain that our approach is conceptually sound and may inspire further research into the flexibility of optimal transport in causal inference. We have detailed the assumptions, limitations, and our scope in the appendix and will emphasize them in the main paper's revision.
>
>
> [1] Lu, Sizhu and Peng Ding. "Flexible sensitivity analysis for causal inference in observational studies subject to unmeasured confounding." (2023).
>
> [2] Zheng, Jia-wen et al. "Copula-based Sensitivity Analysis for Multi-Treatment Causal Inference with Unobserved Confounding." (2021).
>
>
> ---
>
> We hope this response addresses your concerns and elucidates the theoretical foundation of our work. Your feedback is instrumental in refining our paper, and we are open to further dialogue to enhance our contribution.

---

> > ### Comment · Reviewer_VwsR · 2023-08-21
> > **Response**
> >
> > Thanks for detailed clarifiation on my concerns. I am satisfied with your responses. Hence, I would like to raise my score and recommend to accept this paper due to its quality and novelty.

---

### Author Response · Authors · 2023-08-21
**Concluding Response**

Dear Reviewers,

Many thanks for your engagement in the discussion phase. For convenience, we have prepared a concise summary on the final updates according to your suggestions. In the final version, we will

- provide a theoretical analysis of PFOR and its relationship with partial identification [VwsR, MG3J];
- provide a more comprehensive description of related works and include all suggested citations [hjjj];
- revise the paper according to the detailed feedbacks and clarify the strengths of PFOR in the main text, with limitation discussion and toy examples; provide more detailed analysis of MSE issue [MG3J];
- revise the typos and details that are kindly pointed out (e.g., CATE v.s. ITE), reconsider the scope of our title [jCP6].

We are thankful for the feedback and your suggestions on the paper improvement.

---

### Decision · Program_Chairs · 2023-09-21

**Decision:**

Accept (poster)

**Comment:**

Reviewers agreed the paper presents insightful ideas, giving new and deep theoretical results at the intersection of two disparate fields: optimal transport and counterfactual inference, including expanding the use of optimal transport to recently proposed proximal inference approaches. The paper is well-written and the experiments are sound, and contributes to a better understanding of this sub-field of causal effect inference.